# Plasma Bead Entrapped Liposomes as a Potential Drug Delivery System to Combat Fungal Infections

**DOI:** 10.3390/molecules27031105

**Published:** 2022-02-07

**Authors:** Munazza Tamkeen Fatima, Zeyaul Islam, Ejaj Ahmad, Mehboob Hoque, Marriam Yamin

**Affiliations:** 1Interdisciplinary Biotechnology Unit (IBU), Aligarh Muslim University (AMU), Aligarh 202002, India; ejaj_ahmad@rediffmail.com (E.A.); mhbb19huq@gmail.com (M.H.); 2Laboratório Nacional de Biociências (LNBio), Centro Nacional de Pesquisa em Energia e Materiais (CNPEM), Campinas, Sao Paulo 13083-970, Brazil; zislam@hbku.edu.qa; 3Functional and Molecular Biology, Biochemistry, UNICAMP, Campinas, Sao Paulo 13083-970, Brazil; marriam.khalid@yahoo.com

**Keywords:** tissue distribution, amphotericin B, drug–protein interaction, toxicity, plasma beads, sustained release

## Abstract

Fibrin-based systems offer promises in drug and gene delivery as well as tissue engineering. We established earlier a fibrin-based plasma beads (PB) system as an efficient carrier of drugs and antigens. In the present work, attempts were made to further improve its therapeutic efficacy exploiting innovative ideas, including the use of plasma alginate composite matrices, proteolytic inhibitors, cross linkers, and dual entrapment in various liposomal formulations. In vitro efficacy of the different formulations was examined. Pharmacokinetics of the formulations encapsulating Amphotericin B (AmpB), an antifungal compound, were investigated in Swiss albino mice. While administration of the free AmpB led to its rapid elimination (<72 h), PB/liposome-PB systems were significantly effective in sustaining AmpB release in the circulation (>144 h) and its gradual accumulation in the vital organs, also compared to the liposomal formulations alone. Interestingly, the slow release of AmpB from PB was unusual compared to other small molecules in our earlier findings, suggesting strong interaction with plasma proteins. Molecular interaction studies of bovine serum albumin constituting approximately 60% of plasma with AmpB using isothermal titration calorimetry and in silico docking verify these interactions, explaining the slow release of AmpB entrapped in PB alone. The above findings suggest that PB/liposome-PB could be used as safe and effective delivery systems to combat fungal infections in humans.

## 1. Introduction

Fibrin-based formulations are now being extensively used in drug/cell delivery as well as tissue engineering. Their usage in drug therapies includes areas of analgesia, chemotherapy, infection, gene delivery, and regenerative medicine, which benefit from the local delivery of drugs/therapeutics [1,2,3,4,5]. Fibrin formulations, owing to their bio-adhesive properties, have been used as tissue-adherent carriers for the local delivery and slow release of various drugs and growth factors [6,7,8,9,10]. Fibrin–antibiotic mixtures are used in the treatment of infected sites that are difficult to reach with systemically administered antibiotics [11]. Components of the polymer can also be injected and induced to gel in situ [12,13]. Fibrinolytic enzymes such as plasmin gradually break it into small fragments, allowing controlled release of the encapsulated products [14].

Fibrin matrices can be formed under mild experimental conditions in vitro under the influence of intrinsic enzymatic processes and are hence suitable as a carrier of heat-labile drugs, proteins, and enzymes [15]. The porosity of the fibrin mesh formed in vitro is dependent on reaction conditions viz. pH, ionic strength, and the presence of CaCl_2_ suggest that the formulation parameters may be fine-tuned to control the relative availability of the entrapped pharmaceutical [16,17]. Additionally, these can incorporate reactive/fragile molecules such as protein and peptides without serious modifications [15]. Thus, the easy usage and handling of fibrin coupled with its biomimetic proficiencies greatly advocate its possible candidature for a promising drug delivery system [18]. However, the risks of immunological complications arising from the use of commercial fibrin, thrombin, and factor III have been a subject of serious concern [7,19]. While autologous fibrinogen preparations and the use of recombinant human thrombin in place of the bovine enzyme seem to address the risks to a large extent [6], such preparations continue to be expensive. The quick release of the entrapped pharmaceutical accounting to the wide-mesh fibrin matrix, which cannot effectively prevent the diffusion of the antibiotics, is another limitation [20]. Manipulation of the gel, therefore, often forms a prerequisite to ensure the sustained release of the entrapped therapeutic. Our study aims at attaining a continuous and controlled release of pharmaceuticals from fibrin as well as lipid-based preparations. Enhancement of the therapeutic potential of drugs is envisioned via reduced toxicity and dosage frequency, as also suggested by other studies highlighting the interaction of different lipid formulations in reducing toxicity with cellular components, including red blood cells [21].

We have reported a procedure for the preparation of plasma beads (PB) from autologous plasma, necessitating the addition of no exogenous enzymes/proteins, with remarkable potential for sustained drug/antigen delivery [22,23,24]. In this paper, we report the use of plasma-alginate composite matrices, fibrinolytic inhibitors (aprotinin and 6-amino caproic acid) as well as dual delivery via various liposomal formulations to achieve superior sustained release of entrapped molecules in PB. We chose HRP (a small molecule, 44 kDa), blue dextran (a large molecule, 2000 kDa), and AmpB (as a model drug) to investigate the controlled release behavior of the proposed dual delivery system (Appendix A). AmpB was chosen for in vivo delivery owing to its therapeutic potential, as its antifungal properties could be used further to alleviate fungal infections. Plasma and tissue distribution studies of free AmpB, liposome-encapsulated AmpB (Lip–AmpB), PB-entrapped AmpB (PB–AmpB), and AmpB encapsulated in liposome further entrapped in PB (PB–Lip–AmpB) were carried out in mouse model. Formulations were administered intraperitoneally to the different groups of animals. Molecular interactions of the drug with the PB system, including isothermal titration calorimetry and in silico docking, were also investigated.

## 2. Materials and Methods

### 2.1. Materials

Blue dextran, horse radish peroxidase (HRP), AmpB, 6-amino caproic acid, aprotinin, glutaraldehyde, egg phosphatidylcholine (PC), cholesterol, Sephadex G-10, Sepharose 4B, o-dianisidine hydrochloride, and bicinchoninic acid (BCA) were purchased from Sigma Chemical Company, St. Louis, MO, USA. The purity of AmpB was ~80%, HPLC grade. Ethylenediaminetetraaceticacid (EDTA), CaCl_2_, and Triton X-100 were obtained from SISCO Research Laboratory, Mumbai, India. HPLC water, methanol, and chloroform (HPLC grade) was purchased from HiMedia Laboratories Pvt. Ltd., Mumbai, India. Other chemicals used in the study were of analytical grade.

### 2.2. Animals, Collection of Blood, and Isolation of Plasma/Serum

Female Swiss albino mice (6–8 weeks old) weighing 25 ± 2 g were procured from Laboratory Animal Resources, Indian Veterinary Research Institute, Bareilly, India. Rabbits used in the study were outbred and purchased from a local animal supplier. All animals were provided with plenty of water and diet *ad libitum* (Hindustan Lever Ltd., Mumbai, India) throughout the investigation unless mentioned otherwise. Bleeding, injection, and sacrifice of animals were strictly performed following the mandates approved by the Institutional Animal Ethics Committee constituted as per the recommendations of the Committee for the Purpose of Control and Supervision of Experiments on Animals (CPCSEA), Government of India (registration number 332).

### 2.3. Methods

#### 2.3.1. Preparation of PB

PB were prepared by the procedure described by Ahmad et al. [22]. Briefly, 250 µL of plasma was mixed with 50 μL aqueous solution of the substance to be entrapped in 20 mM phosphate-buffered saline (PBS), pH 7.4, and CaCl_2_ (40 mM). Aprotinin and caproic acid were added to achieve a final concentration of 3000 KIU/mL and 10 mg/mL, respectively, in some cases.

Where indicated, PB were crosslinked with glutaraldehyde. For this purpose, appropriate number of beads were suspended in 1.0 mL PBS, and glutaraldehyde solution was added to achieve the final concentration and incubated at 4 °C for 15 min to facilitate crosslinking. The reaction was stopped by adding 500 µL of 0.5 M glycine or methylamine, and the beads washed thrice with normal saline and finally in PBS.

#### 2.3.2. Preparation of Composite Plasma-Alginate Beads

Composite beads of plasma and calcium alginate were prepared. A 2% (*w*/*v*) solution of sodium alginate was prepared by gentle stirring at room temperature for at least 1 h. The alginate solution (100 µL) was mixed in microfuge tubes with 50 µL plasma and 50 µL solution of the pharmaceutical to be entrapped. Finally, appropriate quantities of CaCl_2_ were added, and the plasma-alginate beads were prepared as described above. Increasing the concentration of alginate resulted in rapid gelling and interfered with the formation of beads.

#### 2.3.3. Preparation of Liposomes and Entrapped Pharmaceuticals

The liposomes were prepared essentially by following the published procedure [25]. Briefly, egg PC and cholesterol (7:3 molar ratios or 25 mg egg PC and 5.2 mg cholesterol by weight) were dissolved in 10 mL of chloroform and methanol (9:1 *v*/*v*) and reduced to thin, dry film under N_2_ atmosphere in a round bottom flask of a rotary evaporator. In the case of AmpB, the drug was first dissolved in 2–3 mL methanol and added to the chloroform–methanol mixture with previously dissolved egg PC and cholesterol. The solvents were evaporated in a rotary evaporator to obtain a thin and homogenous lipid film in a rotary flask. The film was then hydrated with an aqueous solution of blue dextran and normal saline in the case of AmpB in a bath sonicator for the formation of liposomes. These were centrifuged thrice at 20,000× g in saline to remove the unentrapped substances. For the preparation of liposomes [25,26] for the entrapment of HRP, the film was hydrated in saline, followed by sonication in a bath-type sonicator (Hwashin Technology Co. Ltd, Seoul, Korea) intermittently for 20 min at 4 °C, under N_2_ atmosphere. The liposomes formed were mixed with an equal volume of HRP (6–10 mg/mL). The mixture was flash frozen and thawed (3 cycles) subsequently and passed through a Sepharose 4B column (bed dimensions 51 × 1.3 cm^2^). Fractions containing liposomes were collected and lyophilized to a minimum volume for use. The amount of encapsulate (blue dextran/AmpB/HRP) was calculated by dissolving 20 µL of the loaded liposomes in 20 mM PBS, pH 7.4, containing 1% Triton X-100. An appropriate volume of the liposomes containing the required amount of encapsulate were added to the plasma mixture for the preparation of beads. Amounts of 100 µg, 1000 µg, and 250 µg of AmpB, blue dextran, and HRP, respectively, were entrapped for in vitro release kinetics experiment. Formulations containing the required amount of Amp B (30 mg/kg weight of the mice) were used for in vivo formulations.

#### 2.3.4. In Vitro Release Kinetics

Release kinetics were followed as described by Ahmad et al. [22]. Briefly, studies on the release of entrapped substances were usually performed with 30–35 plasma beads (necessitating the use of 100 µL plasma, 20 µL containing the entrapped pharmaceutical, and 15 µL of 0.35 M CaCl2). Beads were washed in normal saline, suspended in 1.5 mL of 20 mM PBS, pH 7.4, and incubated at 25 °C (release kinetics should, however, be studied ideally at 37 °C). Released proteins/drugs were quantitated in the PB- free medium. For calculation of the extent of the release, substances present in beads were taken as 100%.

Note: In the case of AmpB entrapment in beads, the drug was pre-dissolved in minimum volume of DMSO, and water was added to achieve the required concentration.

#### 2.3.5. Quantitation of the Entrapped Substances in the Plasma Beads

AmpB and blue dextran were quantitated spectrophotometrically by measuring the absorption at 405 nm [27] or 629 nm [28], respectively. PB with entrapped AmpB/blue dextran was suspended in 1.5 mL of PBS, and the release was monitored in the supernatant. In the case of entrapment in the PB–liposomes system, the supernatants for various time points were further dissolved in 1% (*w*/*v*) Triton X-100. The release was quantitated with the help of a standard curve of the respective molecule in the study. The total drug retained was also calculated by dissolving 5 freshly prepared beads in 1 mL of 0.1 N NaOH to measure the percentage release of the entrapped moiety.

#### 2.3.6. HRP Assay

The HRP assay was performed following the method described by Wititsuwannakul et al. [29]. Three milliliters of assay mixture comprised of 100 µL of 77 mM H_2_O_2_, 100 µL of 25 mM o-dianisidine hydrochloride, 100 µL of the suspending buffer containing HRP, and 2.7 mL of 0.2 M phosphate buffer, pH 6.0. The mixture was incubated for 15 min, at 37 °C. The reaction was stopped by adding 1.0 mL 6.0 N HCl, and absorbance was monitored at 460 nm.

#### 2.3.7. HPLC Analysis of AmpB

AmpB was quantified using HPLC following Nilsson-Ehle et al. [30] with slight modifications, using a Binary HPLC Pump (Waters 1525, Eschborn, Germany) with a Dual λ Absorbance Detector (Waters 2487, Eschborn, Germany). A C18 reversed-phase column (250 mm × 4.6 mm) with the particle size of 5 µm was used. The mobile phase comprised a mixture of 5 mM EDTA and methanol (20:80, *v*/*v*). The flow rate was kept at 1.0 mL/min, and the absorbance of the AmpB was measured at 405 nm. Separate calibration curves of AmpB were prepared after extracting the added drug from plasma, liver, kidney, and spleen tissues. The slope and intercept of the calibration curve were determined by linear regression using the least square method. The extractability of AmpB was determined by adding known amounts of the drug to the tissue homogenate and plasma, followed by its extraction as discussed subsequently [30]. Extraction efficiencies obtained were above 95% in the case of plasma and spleen tissue extracts and above 85% in kidney and liver tissue extracts which were consistent with earlier reports [31,32].

#### 2.3.8. Ampb Analysis in Plasma/Tissues

To one hundred microliters of plasma, 300 µL methanol was added. Liver, kidney, and spleen were washed in PBS, and traces of the buffer solution were removed by blotting with micro wipes. One gram of liver tissue was homogenized using a Remi, Type RQT 127-A homogenizer, Ahmedabad, India and enough cold saline added to obtain volume 2.0 mL, which represents a tissue to liquid ratio of 1:1 (*w*/*v*). In the case of kidney and spleen, the organs were weighed, crushed in saline, and homogenized in a final volume of 1.0 mL. Aliquots of the homogenate were taken and mixed with thrice the volume of methanol and subjected to 60 min incubation at room temperature. The samples were then centrifuged at 10,000× g for 10 min, supernatant collected, and filtered through a 0.45 µm millipore filter. Aliquots (usually 20 µL) of the resultant supernatant were injected into the HPLC system for analysis. The resultant peaks were quantified by a standard of known concentration eluted at the same retention time.

#### 2.3.9. Isothermal Titration Calorimetry (ITC)

ITC experiments were carried out by using a VP-isothermal titration calorimeter (Microcalorimeter, Malvern, PA, USA). BSA, as well as AmpB, was dissolved in 25 mM Tris buffer (pH 7.5). All samples and buffer solutions were degassed at room temperature prior to being used. Experiments were conducted by titrating AmpB (160 μM, in the syringe) into BSA (16 μM, in the cell) at 25 °C. Control experiments were performed to determine the heat of dilution. Each injection of 10 μL with an interval of 3 min was added using a computer-controlled 250 μL microsyringe, and resulting isotherm and heat change parameters were used to calculate thermodynamic parameters for protein-ligand interactions. Control experiments were performed to determine the heat of dilution for each injection by injecting the same amount of BSA or AmpB into buffer alone, and all ITC data were corrected for the heat of dilution of the titrant by subtracting the heats generated by titrating the BSA and AmpB into buffer alone. Data were fitted iteratively until the best fit was obtained. Binding parameters (K_d_ values) were obtained from the fitting of the curve to the best model. Three independent titration experiments were performed, and the averages were taken.

#### 2.3.10. In Silico Docking

BSA crystal structure was obtained from the protein data bank (PDB ID: 4F5S). The downloaded pdb coordinate was edited manually. Only one chain was used for docking, and the water and other heteroatoms were removed from the pdb file. The AmpB sdf file was downloaded from the pubchem database (https://pubchem.ncbi.nlm.nih.gov/) (20 December 2021) and converted to pdb file using OpenBabel software [33]. Molecular docking was performed using the Patch Dock server [34]. The pdb files of BSA were provided as receptor molecule and AmpB as ligand molecule with default values, and RMSD cutoff value of <4.0 Å. Based on the score, top ranked solution was selected for interpretation. The amino acid residues of BSA that are within 4 Å were analyzed for the interaction with AmpB.

## 3. Results

### 3.1. Release of Blue Dextran from PB In Vitro

Blue dextran, a neutral, high molecular mass (2000 kDa) polysaccharide, was selected with a view to studying the retention of neutral large molecules in the PB. On incubation in vitro, the initial rapid release of blue dextran from the beads was followed by a more gradual leakage. Nearly 50% of the entrapped drug was released into the medium in the first 24 h, while the near-complete release was observed in 168 h (Figure 1A). The rate of blue dextran release decreased markedly as a result of crosslinking of the PB with glutaraldehyde. PB treatment with 0.01 or 0.05% (*v*/*v*) glutaraldehyde resulted in the release of 50% of the entrapped blue dextran in 72 and 100 h, respectively. After the completion of incubation (168 h), the release of the blue dextran from the crosslinked beads was low (87% or 77%, respectively, for PB crosslinked with 0.01 and 0.05% glutaraldehyde) as compared to that from uncrosslinked preparations. The leakiness of the beads, however, remained unaffected on the inclusion of calcium alginate in the plasma (Figure 1A).

Encouragingly, release rates of blue dextran from PB decreased significantly when the blue dextran was pre-encapsulated in liposomes. Less than 40% of the liposomally encapsulated blue dextran emerged in the medium in 168 h (Figure 1B), while further treatments with glutaraldehyde (0.05%) led to a significant decrease, releasing only 29% of the entrapped molecule in 168 h.

### 3.2. In Vitro Release of HRP from the PB

Release kinetics from the PB was also investigated using a relatively small glycoenzyme (HRP, 44 kDa). The entrapped HRP unloaded from the PB almost completely in nearly 72 h (Figure 1C). The release curve was biphasic, with over half of the HRP being released during the first 12 h of incubation. As observed earlier with blue dextran, the release rate of HRP from the PB decreased markedly as a result of crosslinking with glutaraldehyde. Crosslinking with 0.01% and 0.05% of the glutaraldehyde caused a reduced release of 83% and 71% HRP, respectively, in 72 h. Encapsulation of the HRP-loaded liposomes in the PB caused a dramatic decrease in the release of the enzyme. Only 21% of the entrapped HRP leaked in 72 h, and only marginal additional release occurred on extending the incubation to 168 h (Figure 1C). Previous reports suggest the use of inhibitors in restricting proteolysis leads to a decrease in the release of substances entrapped in fibrin preparations [35]. The addition of caproic acid (10 mg/mL) and aprotinin (3000 KIU/mL) to the plasma prior to bead preparation, however, did not markedly affect the release rate of HRP (Figure 1C).

### 3.3. Entrapment and In Vitro Release of Entrapped AmpB from Liposomes Further Entrapped in the PB

Efficient entrapment of over 90% was obtained in the case of AmpB (Table 1). Release of the antifungal from PB–Lip–AmpB (Figure 1D) was gradual and remarkably slow, with only about 33% of the entrapped drug leaking out into the medium after 168 h of incubation. Crosslinking with glutaraldehyde resulted in an additional marked decrease in the release rate. The drug release was only 17% in 168 h from beads crosslinked with 0.05% glutaraldehyde.

### 3.4. In Vivo Pharmacokinetic Studies

Improvement in drug pharmacokinetic behavior is the foremost target of drug delivery systems. The present investigation encompasses a pharmacokinetic evaluation of the PB-based formulations in mice along with tissue distribution studies for systemic delivery of AmpB. In an in vivo study in the *Balb*/*c* mice, comparative tissue distribution was performed following intraperitoneal (i.p.) administration of AmpB. Various formulations were administered in separate groups of mice, and the drug was quantitated in blood as well as in other vital organs. The formulations included free AmpB, Lip–AmpB, PB–AmpB, PB–Lip–AmpB. Animals received (30 mg/kg wt) of the drug (free or entrapped) in each formulation. The control group included animals receiving the drug in saline (Sal–AmpB).

### 3.5. AmpB Distribution in Blood Plasma after Administration of the Drug in PB Formulations

Mean plasma AmpB concentrations–time curves after i.p. administration are shown in Figure 2A. The plasma concentration of AmpB was maximum (2.43 µg/mL) at 4 h after the administration of the free drug; however, the level decreased rapidly and became undetectable at 72 h. Plasma levels of drugs of animals receiving Lip–AmpB decreased gradually from the initial 2.14 µg/mL observed at 4 h to non-detectable levels at the end of 144 h. Animals receiving PB–AmpB and PB–Lip–AmpB exhibited maximum plasma drug concentrations of 1.5 µg/mL at 24 h, but the drug level declined very gradually with almost 50 percent (0.66 and 0.7 µg/mL, respectively) of the drug still present in the circulation after 144 h.

### 3.6. AmpB Profiles in Liver

AmpB analysis in the liver shows the maximum concentration of the drug in the group receiving the free drug at 24 h, with the level of AmpB reaching about 40 µg/g of the tissue (Figure 2B). It then registered a rapid decline and reached 21 µg/g and 9 µg/g at 48 h and 72 h, respectively. In the animals of the Lip–AmpB group, maximum drug levels were also reached at 24 h, but the levels were significantly lower compared to the control group, and the subsequent decrease was also more gradual. At the end of 144 h, the amount of drug retained in the livers of animals of the Lip–AmpB group was moderately higher than that of the control. In the case of animals receiving PB–AmpB or PB–Lip–AmpB, the hepatic drug level was quite low at 24 h (5 µg/g and 4 µg/g, respectively) but continued to increase gradually up to 144 h. The level of the drug after 144 h was 17 µg/g and 14 µg/g in PB–AmpB or PB–Lip–AmpB groups, respectively.

### 3.7. AmpB Profiles in Spleen

The level of AmpB in the spleens of mice receiving free AmpB was as high as 71 µg/g at 24 h after the administration of the drug (Figure 2C), decreased gradually, and reached 27 µg/g at the end of 144 h. In the Lip–AmpB group, the drug levels were 43 µg/g at 24 h and 19 µg/g at 144 h. On the other hand, in the groups of animals receiving PB–AmpB or PB–Lip–AmpB, the drug levels were low at 24 h (i.e., 15.6 and 16.7 µg/g respectively), but they gradually rose to 35 μg/g and 40 µg/g, respectively, at 144 h.

### 3.8. AmpB Profiles in Kidney

The distribution of AmpB levels in the kidneys of animals receiving various formulations of the drug was also studied, and the pattern obtained was comparable to that observed in the liver. Maximum drug levels were recorded in mice given free AmpB (19.7 µg/g) at 24 h but decreased rapidly to 4.8 µg/g by 144 h (Figure 2D). The maximum level of AmpB observed in mice of the Lip–AmpB group after 24 was low (12.8 µg/g) as compared to the control group. Kidney drug levels in this group also decreased and became very low (4.75 µg/g) at the end of 144 h. In animals receiving AmpB entrapped in PB (PB–AmpB) or after encapsulation in liposomes (PB–lip–AmpB), the detectable drug levels were 2.3 µg/g and 2.7µg/g at 24 h, followed by a subsequent gradual increase. The kidney level of the drug in PB–AmpB or PB–Lip–AmpB was 10.7 µg/g and 10.2 µg/g, respectively, by 144 h.

### 3.9. Interaction of AmpB with BSA

The interaction of AmpB with BSA was probed by ITC and in silico docking methods. A high dissociation constant (K_d_) of 3.58 ± 1.6 µM was obtained by fitting the data to single-mode interaction (Figure 3A), reflecting a strong drug–protein interaction. In silico docking studies further validates the biophysical result. The monomeric BSA (Figure 3B) has a multi-domain architecture (three domains), giving it an overall heart shape structure. The monomeric BSA, along with AmpB was used for computational docking to analyze the binding of AmpB to the BSA. Clearly, the drug binds to the cavities of domain 1 and domain 2 (Figure 3C). The domain 2 of BSA, based on hydrogen and hydrophobic interaction, was suggestive of being the binding pocket of AmpB [36]. AmpB being hydrophobic, interacts mainly with the hydrophobic residues of the BSA (tryptophan and tyrosine) (Figure 3D).

## 4. Discussion

Beads prepared from plasma show remarkable potential as efficient delivery systems [22,23,24]. The risk of immunological complications could most possibly be minimized since the formulations utilize autologous plasma without the need for any foreign protein/enzyme. Total biodegradability and biocompatibility, as well as ease of preparation using a simple laboratory setup, are the other advantages. Previous attempts of using fibrinogen from autologous blood have also been made [38,39,40].

Both large and small molecules like blue dextran and HRP could be readily entrapped in PB with high yields, as reported in our earlier work [41]. The beads, however, readily released blue dextran (Figure 1A) and HRP (Figure 1C), suggesting their porous nature. Glutaraldehyde treatment caused a remarkable decrease in the overall release rates (Figure 1A–D). Other reports also suggest that the porosity of the fibrin matrices can be decreased on crosslinking with glutaraldehyde [42,43] and other bifunctional reagents like 1-ethyl-3- (1–3 dimethyl aminopropyl) carbodiamide (EDC) and genipin [44]. Since blue dextran does not contain amino groups that can be crosslinked with the dialdehyde to the plasma proteins, the lowered rate of their release from the PB can be attributed to a decrease in pore size of the protein network upon crosslinking.

Dual entrapment has also been a useful strategy in manipulating the controlled release of pharmaceuticals from leaky/porous delivery systems and decreasing their release significantly or increasing the overall efficiency of the proposed system [43,45,46,47]. Pre-entrapment in multilamellar liposomes and crosslinking provides a significant reduction in the release rate (Figure 1B). The liposomes were prepared using PC and cholesterol and hence offered no groups to react with the amino group of bifunctional glutaraldehyde. A decrease in the release of the liposome entrapped blue dextran is, therefore, the result of concentration-dependent crosslinking of the fibrin network rather than the covalent attachment of the vesicles to the PB proteins.

Release time of drugs in vitro has been reported to be delayed remarkably from sodium alginate-fibrin composite supports as compared to alginate alone [48,49]. The inclusion of sodium alginate in plasma beads in our studies, however, failed to produce any significant effect on the release kinetics of large molecules as blue dextran (Figure 1A) and small molecules such as AmpB (data not included). This may be due to insufficient concentration of fibrinogen, unable to reduce the porosity of the matrix, or interference caused by the alginate itself to effectively form the fibrin clot.

Rapid release of HRP from the beads (Figure 1C) may be accredited to its relatively lower molecular weight (44 kDa) and high-water solubility. HRP has several free amino groups (6 lysine residues), which are likely to be crosslinked to the PB proteins during treatment with glutaraldehyde [50]. The decrease in the observed release rate of the enzyme is, therefore, the result both of covalent coupling of HRP to plasma proteins as well as covalent crosslinking of some of the fibrin network decreasing the pore size of the PB. HRP was entrapped in liposomes efficiently with an entrapment yield of ~50% [41], utilizing a slightly different procedure (detailed in methods). Pre-encapsulation of the enzyme in liposomes prior to entrapment in plasma beads, however, decreased the release of HRP remarkably (~80%). The effect of pre-entrapment in the liposomes was apparently more marked in the case of HRP as compared to the release of blue dextran due to large differences in the dimensions and charge on the vesicles.

High entrapment efficiencies of AmpB in liposomes have been accomplished (Table 1), which may be attributed to the chemical nature of the drug [51]. AmpB is an amphipathic drug containing several free hydroxyl groups and a long hydrocarbon polyene chain that facilitates its entrapment both in the aqueous hydrophilic compartments as well as intercalation in the hydrophobic region of the lipid bilayers. PB–Lip–AmpB formulations did not significantly decrease the release rate of AmpB from the PB (Figure 1D) as observed with blue dextran and HRP. This could be due to the high binding of the drug to matrix proteins, which possibly forms the critical determining factor in the overall release.

Earlier studies on the release of antibiotics such as Ampicillin, Carbenicillin, Ceftazydime, Clindamycin, Gentamycin, Cefotaxim, Mezlocillin, and Tobramycin from fibrin clots have revealed near the complete release of the drugs by 96 h [11]. The release occurred faster in cases of Ampicillin (48 h) and Clindamycin (72 h). Another study also suggested almost a complete release of the antibiotics Gentamycin, Neomycin, and Polymyxin E from the fibrin clots within 72 h [12]. AmpB release from the PB revealed a surprisingly different kinetic profile, with not more than 25% of the entrapped drug leaking out in 100 h [22] against almost a total leakage observed in the case of most other drugs [11,12,22]. The observed slow release of AmpB from the PB may be attributed to its high affinity for plasma proteins, especially to human serum albumin and human α_1_-acid glycoprotein [52] that are the constituents of plasma beads. Raza et al. proposed the interactions of AmpB and BSA using multi-spectroscopic techniques, including fluorescence spectroscopy, circular dichroism, Fourier transform infrared spectroscopy, and in silico molecular simulation studies [36]. The AmpB–BSA complex formation was studied by ITC and docking analysis in the present study. K_d_ values, in the micromolar range, suggested a tight molecular interaction between the drug and the protein. A similar dissociation constant (K_d_ value of 1.25 µM) was reported recently by using the fluorescence quenching titration method [36]. In silico studies predicted the binding interface of AmpB in the domains I and II of BSA, although crystal structure determination is required to determine the exact interface of the drug.

Fungal infections pose a major difficulty in the management of immunocompromised patients [53]. Despite the well-known toxicities and treatment letdowns, AmpB has been widely used for the treatment of these infections [31]. The newer triazoles, such as posaconazole and isuconazole, as well as AmBisome (liposomal amphotericin B), are also now used for treating various mold infections. Entrapping AmpB into multi/unilamellar liposomes [54] or binding to the other substances [52] reduces its toxicity to host cells. The plasma/tissue distributions results of the injected liposomized drug in blood/vital organs suggested favorable pharmacokinetic behavior [55]. Thus, the use of less toxic formulations has been envisioned to address the problem of toxicity without compromising the drug dosage.

AmpB, when administered in free form, is rapidly taken up by recticoloendothelial system and cleared from the circulation with little drug remaining in blood after 72 h (Figure 2A). The apparent circulating half-life (t_1/2_) in the case of animals receiving free AmpB was less than 20 h, while it was about 68 h in the case of those administered the liposome-encapsulated drug. Gondal et al. [54] observed similar retention of liposome-encapsulated AmpB in circulation after i.v. administration in mice. Interestingly, administration of the drug entrapped either directly in the PB or after its pre-encapsulation in liposomes revealed remarkably altered pharmocokinetics. In both cases, the drug exhibited a t_1/2_ of about 120 h, indicating a far gradual clearance from the plasma or elimination by recticuloendothelial system. The comparable pharmacokinetic behavior of the drug in these two formulations is corroborated by the in vitro release rate of PB. An earlier study from this laboratory has shown similar far gradual in vitro and in vivo release of cefotaxime entrapped in PB [22]. The high apparent half-lives of the drug in the animals receiving the PB formulations is due to the sustained release of the free and liposomized drug facilitating their retention in the plasma for longer durations. The findings suggest that the formulation could be used in lower doses or less frequent drug administration, thereby reducing the toxicity of AmpB, particularly to the erythrocytes [56] and nephrocytes [57]. Rapid clearance of plasma concentration as in the case of animals receiving the free drug is indicative of its rapid absorption and distribution in organs such as the spleen, kidney, and liver, as evident from tissue distribution studies, indicating its rapid uptake by the reticuloendothelial cell-rich organs [54].

Tissue distribution studies of formulations PB/PB–lip–AmpB in mice suggest that the administered AmpB preferentially reached the spleen followed by liver and kidney. Mice receiving free AmpB had remarkably high initial levels of the drug in all these organs compared to the animals receiving the liposomized drug. On the other hand, for the animals receiving PB/PB–lip–AmpB, the AmpB level in these organs continued to increase gradually, suggesting the gradual release of the drug from the PB into the blood and then to various organs. Liposomized AmpB also released gradually into the circulation and subsequently reached the organs studied. Liposome encapsulation improves therapeutic index [58,59], lowers toxicity, and facilitates the use of higher AmpB [60] even when administered intravenously into the mice [54].

In conclusion, the present study was focused on developing a safe, sustained, and effective fibrin-based system to deliver macro and micro molecules. Encapsulation in liposomes significantly reduced the leakage of entrapped molecules. However, formulations Lip–PB/PB were highly effective in ensuring a sustained release of the entrapped AmpB. This reduces the possibility of toxicity by restricting rapid delivery into circulation and the achieved sustainability of the drug in the organs in the required therapeutic range, also minimizing the dosage frequency.

## Figures and Tables

**Figure 1 molecules-27-01105-f001:**
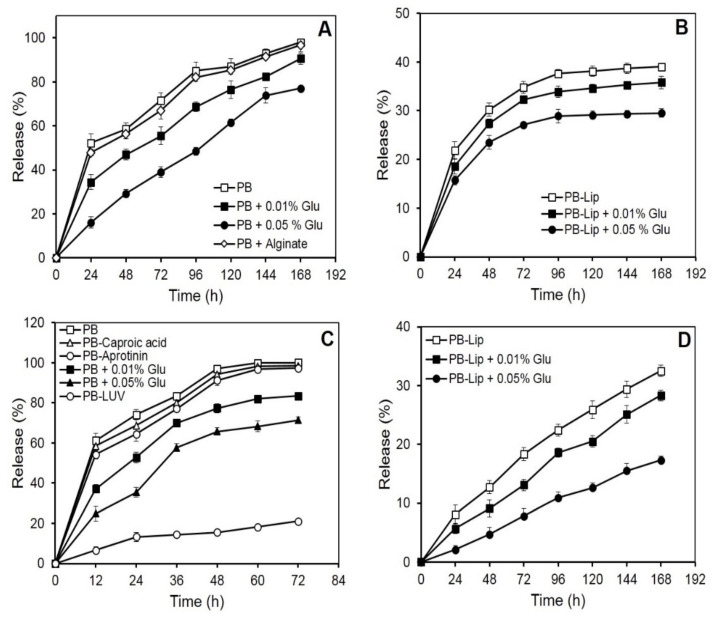
In vitro release of macromolecules from PB and liposomes entrapped in PB formulations. The samples were incubated in PBS at 25 °C, and release kinetics was determined spectrophotometrically as described under Materials and Methods. Each value represents mean ± SD of four determinations. Panel (**A**): release of blue dextran (1000 µg) entrapped in PB. Panel (**B**): release of blue dextran (1000 µg) from liposomes entrapped in PB formulations. Panel (**C**): release of HRP (250 µg) from different PB formulations. Panel (**D**): release of AmpB (100 µg) from liposomes entrapped in PB formulations.

**Figure 2 molecules-27-01105-f002:**
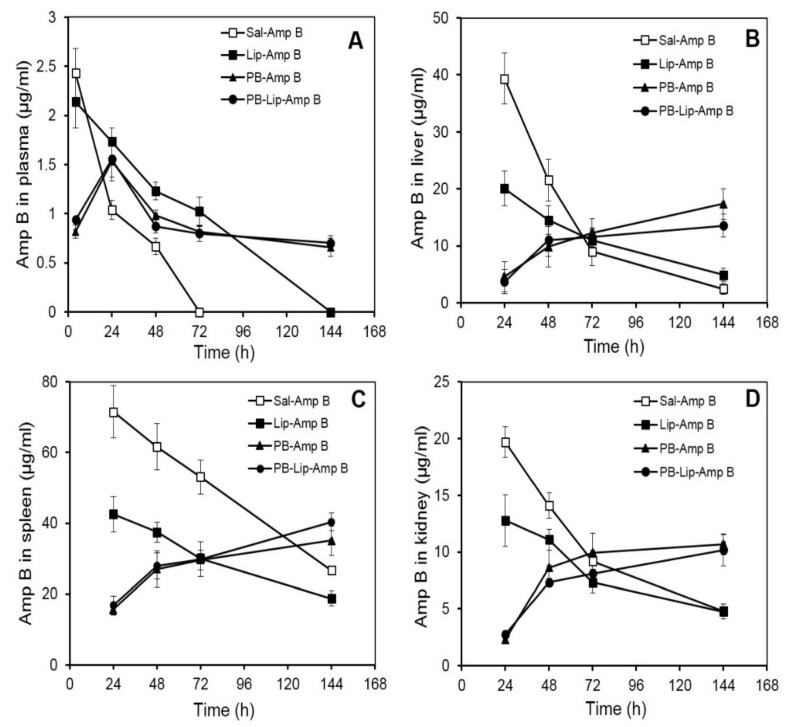
AmpB distribution in blood and organs of *Balb*/*C* mice after administration of the drug in various formulations. AmpB concentrations in the plasma (**A**), liver (**B**), spleen (**C**), and kidney (**D**) of mice after administered the formulations intraperitoneally were determined as described under Methods. Animals received 30 mg/kg body weight of the drug. The animals were bled after various time intervals, and plasma, liver, spleen, and kidney concentration of the antifungal was determined by reverse-phase HPLC as described. Each value represents the mean ± SD of four determinations.

**Figure 3 molecules-27-01105-f003:**
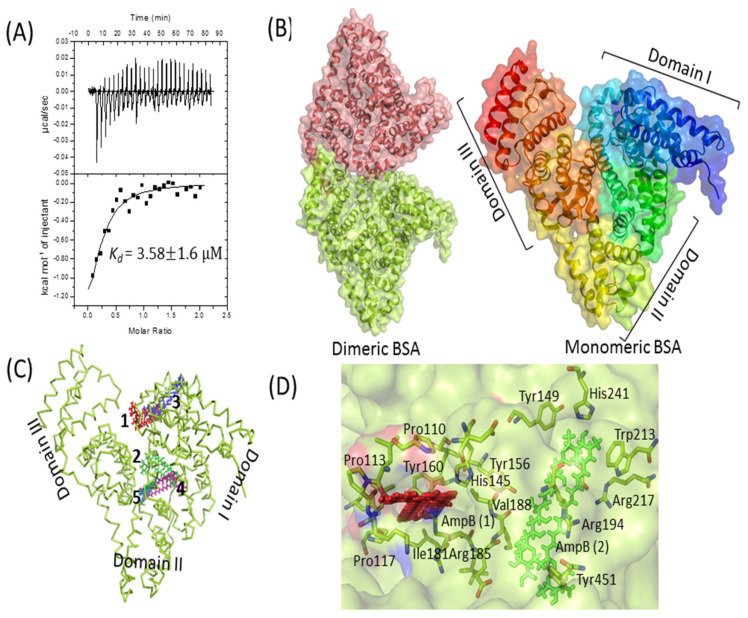
BSA–AmpB interaction studies. (**A**) Isothermal titration calorimetry analysis of the interaction between BSA and AmpB. Data are plotted as heat signal (μJ/sec) versus time (min) in the upper panel and in the bottom panels; the integrated heat responses per injection are plotted vs. molar ratio. The solid line represents the best fit of the data to a single-site binding model. Titration of BSA (16 μM, in the cell) with AmpB (160 μM, in the syringe) was performed at 25 °C. (**B**) Structure of dimeric (left side) and heart-shaped monomeric (right side) BSA, represented as cartoon with transparent surface. BSA pdb coordinate was downloaded from protein data bank (PDB ID: 4F5S). (**C**) Top 5 docked positions of AmpB (shown as numbers 1–5 in different colors) are represented as sticks in the cavities of BSA. (**D**) The binding interaction of AmpB is represented as red and green color with the surrounding residues from the BSA. All the structures were generated by PyMOL [37].

**Table 1 molecules-27-01105-t001:** Entrapment of AmpB in liposomes.

AmpB Added (mg)	AmpB Entrapped mg/mL Packed Liposomes	Entrapment (%)
1.0	3.89 ± 0.27	96
2.0	7.56 ± 0.21	94
3.0	10.94 ± 0.19	91
4.0	14.47 ± 0.47	90
6.0	15.9 ± 0.34	67

## Data Availability

This research is part of the research thesis program and is partly available online.

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
