# Peer review of "Plasma Bead Entrapped Liposomes as a Potential Drug Delivery System to Combat Fungal Infections"

_molecules, 2022, doi:10.3390/molecules27031105_

Round 1
Reviewer 1 Report
The submitted manuscript deals with development and evaluation of plasma beads containing liposomally-encapsulated antifungal drug. The topic is interesting; however, there are deficiencies that should be adequately addressed as commented below.
Abstract: Some of the major results should be presented also by numerical data.
Line 28: please check and re-write “unsual”
Introduction: the authors should provide physicochemical properties of all entrapped compounds, i.e., data for amphotericin B (Amp B: solubility, log P, Mw), as well as for dextran and HRP.
It would be clearer to add (at the end of Introduction) the intended route of the investigated delivery system. Moreover, the reasons (advantages) for encapsulating amphotericin B (Amp B) should be shortly explained.
Materials: phosphatidylcholine is one word, (not 2 words), similarly rewrite ethylene diamine tetra acetic acid to ethylenediaminetetraacetic acid. HPLC is always written with upper case letter, please re-write hplc (line 83). Later in text triton-X 100 (Triton X-100).
Lines 115-116: check the font size.
Why were different morphological types of liposomes prepared (MLVs and LUVs), what were the reasons to use different types of liposomes? This should be explained/discussed in manuscript. The amount of the phospholipid (PC) or total lipid should be indicated (not only the ratio of liposomes’ bilayers constituents). Moreover, what were the amounts of Amp B, dextran and HRP used during liposomes preparation? (Section 2.3.3.)
The authors should be more precise and indicate the exact volumes or masses of liposomes and plasma mixture during preparation of beads.
How could be the authors sure that LUVs are formed? Lamelarity of the liposomes was not tested.
I could not find data on the sizes (mean diameters, polydispersity) of the different liposomes and their zeta potentials? There is only data for LUVs that they were 100-200 nm (in text). Liposomes’ size determination was not indicated in experimental section. Which technique was used for LUVs size measurement? Have the authors checked the mean diameters of MLVs? The size is important parameter for liposomes as delivery system and must be provided.
What was the size of plasma beads and plasma beads with liposomes?
There are no results on encapsulation efficiency for HRP and dextran, only for Amp B.
Amp B is not soluble in water. Centrifugation method is therefore not suitable for separation of the liposomally-entrapped drug as the non-encapsulated drug will precipitate together with liposomes.
Why were Egg PC and cholesterol selected for liposomes preparation? Shortly discuss the reasons for selecting egg PC as major bilayer building compound. Why were not negatively charged phospholipids used at certain ratio as liposomes' membrane constituents?
In vitro release studies are not described in experimental section (I could not find). Since significant part of the study is based on in vitro release behavior of the encapsulated compounds from the different types of liposomes, the experimental set up (whole procedure) should be presented including the method used, volume or mass of liposomes, volume of release media, procedure, etc..
Why were in vitro release studies performed at 25 °C? (Figure 1 legend), why not at 37 °C? There is no sense to study release of the liposomally-encapsulated drug (compounds) at room conditions.
The term liposomized is not appropriate (line 218); instead of this term, I suggest liposomally-encapsulated or entrapped dextran.
Author Response
Response to Reviewer1 Comments
Concern 1. Abstract: Some of the major results should be presented also by numerical data
Response 1. Thank you for the suggestion, we have added numerical data in the abstract.
Concern 2. Line 28: please check and re-write “unsual”.
Response 2. We have made the correction
Concern 3. the authors should provide physicochemical properties of all entrapped compounds, i.e., data for amphotericin B (Amp B: solubility, log P, Mw), as well as for dextran and HRP.
Response 3. Thank you for your valid suggestion. We have added a supplementary Table as Table S1.
Concern 4. It would be clearer to add (at the end of Introduction) the intended route of the investigated delivery system.
Response 4. Thank you for pointing this out. We have mentioned the route of administration in the text.
Concern 5. Moreover, the reasons (advantages) for encapsulating amphotericin B (Amp B) should be shortly explained.
Response 5. We have added detailed text on this in the introduction section. I hope it is okay now.
Concern 6. Materials: phosphatidylcholine is one word, (not 2 words), similarly rewrite ethylene diamine tetra acetic acid to ethylenediaminetetraacetic acid. HPLC is always written with upper case letter, please re-write hplc (line 83). Later in text triton-X 100 (Triton X-100). Lines 115-116: check the font size.
Response 6. Thankyou for point the errors. We have corrected all of them.
Concern 7. Why were different morphological types of liposomes prepared (MLVs and LUVs), what were the reasons to use different types of liposomes? This should be explained/discussed in manuscript.
Response 7. The different types of liposomes were chosen based on the drug entrapment. With AmpB and blue dextran, the encapsulation was not an issue and an entrapment efficiency of > 95 % in case of Amp B (Table 1) and over 65% in blue dextran (Fatima et al., 2018) was obtained with the basic multilamellar vesicles. However, due to the low entrapment of HRP in MLVs (data not included), we used large unilamellar vesicles to obtain better entrapment (~54 % with LUVs, Fatima et al., 2018, reference also provided in the text).
In addition, the main idea was to exploit the utility of fibrin as a delivery system due to the several advantages it offers, as mentioned in the introduction and discussion sections. Albeit, the porous nature of the fibrin network caused a fast release of entrapped drugs (except AmpB) and other molecules of interest. To achieve the best, i.e., utilizing its potential and overcoming the drawback, an alternative approach (double encapsulation) has been used by encapsulating liposomes, which are also prone to quick elimination by recticulo-endothelial system from the circulation. Thus, although we tried to best encapsulate the entrapped substance and used MLVs and LUVs as the case maybe, our focus remained on increasing the retention in the plasma beads which has potential to be useful for human use and in personalized drug delivery approaches.
We have added this briefly in the text.
Concern 8. The amount of the phospholipid (PC) or total lipid should be indicated (not only the ratio of liposomes’ bilayers constituents.
Response 8. We have incorporated the amounts used in the text.
Concern 9. Moreover, what were the amounts of Amp B, dextran and HRP used during liposomes preparation? (Section 2.3.3.)
Response 9. An appropriate volume of the MLVs/LUVs containing the required amount of encapsulate were added to the plasma mixture for the preparation of beads. An amount of 100 µg, 1000 µg and 250 µg of AmpB, blue dextran and HRP respectively for in vitro release kinetics experiment. Formulations containing the required amount of Amp B (30mg/kg weight of mice) were used for in vivo formulations (added in the manuscript).
Concern 10. The authors should be more precise and indicate the exact volumes or masses of liposomes and plasma mixture during preparation of beads.
Response 10. Thank you for your suggestion. We have added this in the manuscript.
Concern 11 & 12. How could be the authors sure that LUVs are formed? Lamellarity of the liposomes was not tested.
I could not find data on the sizes (mean diameters, polydispersity) of the different liposomes and their zeta potentials? There is only data for LUVs that they were 100-200 nm (in text). Liposomes’ size determination was not indicated in experimental section. Which technique was used for LUVs size measurement? Have the authors checked the mean diameters of MLVs? The size is important parameter for liposomes as delivery system and must be provided.
Response 11 & 12.
The PC liposome was prepared by following the procedure optimized in our laboratory earlier and the formulations were characterized previously (Reference provided in the text). Therefore, effort was not made to characterize the liposomes, as the preparation was not unique in this study. The study was focused on the use of fibrin-based plasma beaded system as a novel delivery vehicle. Liposomes were investigated as a system to achieve sustained release via dual entrapment.
Concern 13. What was the size of plasma beads and plasma beads with liposomes?
Response 13. The size of the beads could be manipulated varying the total volume of the plasma
as discussed in detail in our previous paper where we optimized the plasma beads (Ahmad et al., 2011). With the addition of liposomes also, we could regulate the size of the beads.
Concern 14. There are no results on encapsulation efficiency for HRP and dextran, only for AmpB.
Response 14. We have reported the entrapment efficiency of HRP and blue dextran in our previous work and the reference has been provided in the text.
Concern 15. Amp B is not soluble in water. Centrifugation method is therefore not suitable for separation of the liposomally-entrapped drug as the non-encapsulated drug will precipitate together with liposomes.
Response 15.
We solubilized AmpB in minimum volume of DMSO and then water was added to get required concentration of drug (added to the text). The solution was centrifuged to obtain the MLVs while the unentrapped drug remaining in the supernatant was removed. Further, free drug was removed by washing. The procedure was repeated until the drug was not present in supernatant.
Concern 16. Why were Egg PC and cholesterol selected for liposomes preparation? Shortly discuss the reasons for selecting egg PC as major bilayer building compound. Why were not negatively charged phospholipids used at certain ratio as liposomes' membrane constituents?
Response 16. Egg PC and cholesterol are more often used to prepare liposomes. PC is cheap and can also be extracted from eggs in the laboratory. Egg PC was used because of its neutral nature as our purpose was to create a universal system for the entrapment of drugs/molecules in liposomes and further entrap it in the plasma beads, to create a robust dual entrapment system. This was intended to overcome the leakage from the beads that were somewhat porous, as also discussed in the text, rather than analyzing other parameters of the liposomes.
Concern 17. In vitro release studies are not described in experimental section (I could not find). Since significant part of the study is based on in vitro release behavior of the encapsulated compounds from the different types of liposomes, the experimental set up (whole procedure) should be presented including the method used, volume or mass of liposomes, volume of release media, procedure, etc.
Response 17.
I am grateful to the erudite reviewer for pointing out the valid issue. We have added a new section (2.3.4) including all the details.
Concern 18. Why were in vitro release studies performed at 25 °C? (Figure 1 legend), why not at 37 °C? There is no sense to study release of the liposomally-encapsulated drug (compounds) at room conditions.
Response 18. The query of reviewer is valid. It is true that the physiological temperature is 37°C. Therefore, in vitro study should be performed at this temperature. However, In vitro release studies are often performed at ambient /room temperature (25 °C) to evaluate release behavior (Ahmad et al., 2011). As the temperature raised to 37 °C, the release would be a little faster due to the influence of temperature; however, the pattern of release would in large be similar [(Shi et al., Drug Release of pH/Temperature-Responsive Calcium Alginate/Poly(N-isopropylacrylamide) Semi-IPN Beads. Macromolecular Bioscience, 2006, 6(5): 358-63;
Werzer, et al., Drug Release from Thin Films Encapsulated by a temperature responsive Hydrogel. Soft Matter 15 (8)]
Concern 19: The term liposomized is not appropriate (line 218); instead of this term, I suggest liposomally-encapsulated or entrapped dextran.
Response 19. We have incorporated the reviewer's suggestion. Thank you.

Reviewer 2 Report
well designed and nicely presented work. Appreciate the authors.
Author Response
No comments.
Reviewer 3 Report
Research to reduce the toxicity of drugs is one of the most promising and significant areas of modern science. Amphotericin-B, as an antimycotic agent, is one of the most effective and at the same time toxic drugs.
The paper describes in detail the experimental part, but there are no data on:
1. Distribution of Amphotericin-B in liposomes.
2. Has amphotericin-b retained its fungicidal properties? It is necessary to check the effectiveness of the presented associates, at least on SAP enzymes.
3. In manuscript is no characterization of the resulting delivery systems. It would be very interesting if the liposomes structure can be characterized via cryo-TEM.
4. Loading efficiency (DLE/DLC data) of AmpB is not addressed in the manuscript.
5. Cytotoxicity should be evaluated in normal cell lines as well. The IC50 values of various formulations should be reported.
In the introductory part after using 60 links. However, only 12 of them belong to modern studies (over the last 5 years). Of these, only one is devoted to the encapsulation of amphotericin-b. Are the authors confident in the relevance of the study?
Author Response
Response to Reviewer3 Comments
Concern 1. Distribution of Amphotericin-B in liposomes.
Response 1. We have described the mechanism of Amp B distribution in liposome in the discussion section (line 377-380).
Concern 2. Has amphotericin-b retained its fungicidal properties? It is necessary to check the effectiveness of the presented associates, at least on SAP enzymes.
Response 2. Nanomedicine is an area of intensive research, and various drug encapsulated nanocarriers are commercially available in the treatment of cancer, such as DaunoXome, Doxil and Myocet (Ahmad et al., 2021). Moreover, Amp B encapsulated liposome (AmBisome®; LAmB) are commercially available to treat fungal infections. This validates that encapsulation of Amp B into liposomes does not affect the activity of the drugs. Also, we followed published procedure to encapsulate Amp B in the lipid formulations. Therefore, the activity of the Amp B is not likely to be lost.
We have checked the activity of the AmpB encapsulated in the plasma beads and evaluated their antifungal activity on the fungal growth inhibition. We observed fungal growth inhibition similar to that in free form of commercial Amp B (unpublished work).
Concern 3. In manuscript is no characterization of the resulting delivery systems. It would be very interesting if the liposomes structure can be characterized via cryo-TEM.
Response 3. Thank you for your suggestion. However, under the current Covid-19 situation, all the central facilities are almost closed. It is difficult to execute the same within a very short time of 9-10 days. We sincerely apologize for the same.
Concern 4. Loading efficiency (DLE/DLC data) of AmpB is not addressed in the manuscript.
Response 4. We have included this in the result section. It had been discussed already in the discussion (lines 394-398).
Concern 5. Cytotoxicity should be evaluated in normal cell lines as well. The IC50 values of various formulations should be reported.
Response 5: The suggestion from the erudite reviewer is valid. At the same time, we apologize for not being able to perform the experiment due to the lack of facilities as the lab is currently following the work from home protocol.
Also, the very idea of the proposed delivery system is to use autologous plasma and entrap the molecule of interest without introducing any chemical modification or exposure to the entrapped substance to lose its properties. It uses both plasma (very natural) and liposomal (extensively used and known for drug entrapment purposes) utilizing a simple protocol which does not necessitate addition of any external agent that could possibly pose a harm to any moeity entrapped. The current work focusses on the use of plasma beads and entrapped molecule as a system of sustained delivery.
Concern 6: In the introductory part after using 60 links. However, only 12 of them belong to modern studies (over the last 5 years). Of these, only one is devoted to the encapsulation of amphotericin-b. Are the authors confident in the relevance of the study?
Response 6:
Liposomal formulation of Amp B is commercially available and enormous research from our laboratory/department has been performed (Owais et al.). However, the entrapment of pharmaceutical in the plasma bead is the innovation of our laboratory. We have unraveled the usefulness of the system in in vitro and in vivo release of various drugs (Ahmad et al., 2011). We have also evaluated the usefulness of the plasma bead in the vaccine development (Ahmad et al., 2012). Therefore, we are definitely confident in our present study to evaluate its efficacy in encapsulation of anti-fungal drug as well as the in vitro and in vivo release. Our focus was to develop plasma bead-liposome system; however, the unique binding behaviour of AmpB with plasma beads led to its outstanding sustained release behavior in vitro. Thus, we were intrigued to further investigate its in vivo delivery potential and molecular interaction studies.

Reviewer 4 Report
The design of delivery system against fungal infections is interested, while there are too many flaws. (1) Language needs to be overhauled; (2) No characterization details to validate the successful preparation of the drug carrier, such as TEM. (3) The title is “Liposome-plasma bead as a safe and sustained delivery system against fungal infections”, while no experiments show the inhibition dynamic to fungal infections.Author Response
Response to Reviewer4 Comments
Concern 1: Language needs to be overhauled
Response 1: We have improved the language.
Concern 2: No characterization details to validate the successful preparation of the drug carrier, such as TEM.
Response 2: The suggestion from the erudite reviewer is valid; however, in the current pandemic situation it is difficult to perform the experiment in the short time provided. We sincerely apologize for the same
Concern 3: The title is “Liposome-plasma bead as a safe and sustained delivery system against fungal infections”, while no experiments show the inhibition dynamic to fungal infections.
Response 3: The idea of the proposed delivery system involves the to use of autologous plasma and entrap the molecule of interest without introducing any chemical modification or exposure to the entrapped pharmaceuticlas to lose its activuty. It uses both plasma (very natural) and liposomal (extensively used, safe and known for drug entrapment purposes commercially) utilizing a simple protocol which does not necessitate addition of any external agent that could possibly pose a harm to any moeity entrapped. Thus, the use of autologous plasma makes it a safe system for delivery. We observed a sustained release of entrapped AmpB and the pharmacokinetics suggest availability of the drug in the therapeutic window for a longer time, also avoiding its rapid entry or elimination from the circulation. This suggests its usefulness as a potential delivery system against fungal infections
We have however modified the title including 'potential drug delivery system' to better suit the concerns.
'Plasma bead entrapped liposomes as a potential drug delivery system to combat fungal infections'

Round 2
Reviewer 1 Report
Amp B is a poorly soluble drug, and it should be added together with phospholipids and cholesterol during liposomes preparation, not during film hydration phase as indicated by the authors. These are the basics of liposomes and liposomes preparation. Moreover, the authors must indicate the amount of Amp B added to form the Amp-liposomes. The authors indicated the amounts of the drug and markers for in vitro release and in vivo studies.
Since the drug is not soluble in water, ultracentrifugation is not appropriate method for separation of the non-entrapped drug as the drug will precipitate together with the liposomes. Instead, gel-filtration method based on the use of Sepharose CL-4B or minicolumn centrifugation method (Sephadex) should be used. These methods are accurate for all the drugs, while ultracentrifugation is suitable only for hydrophilic drugs.
Although AmpB is soluble in DMSO, after addition of water (higher amount), it will start to precipitate. The encapsulation efficiency is, therefore, false positive. It could be so high but only if the drug is added together with the lipid phase and by selecting the appropriate lipids (please check the composition of commercially available amphotericin B liposomes).
Moreover, the authors should be aware that certain amount of DMSO is still present in liposomes. Such formulation is not appropriate for further human use.
I suggest the authors to prepare the Amp B-liposomes with adding the drug together with the phospholipid and cholesterol, which is a regular way of preparation liposomes with lipophilic drugs. The encapsulation efficiency determination should be checked again, but with using gel-column filtration/chromatography method using Sepharose CL-4B or minicolumn centrifugation method (using appropriate Sephadex).
The authors’ explanation why were in vitro release experiments performed at 25C, not at 37C, must be denoted in the manuscript. The authors must indicate in the revised manuscript that 25 C is not appropriate temperature.
Despite of the cited reference, the lamellarity of the liposomes was not checked and TEM figures are not provided. The used terms LUVs and MLVs are hence not appropriate. Instead, only term liposomes, but indicating the entrapped compound, e.g., Amp B-liposomes, should be used. Please re-write throughout the manuscript.
In response to the reviewer, the authors must specify exactly what change is and where the change is made in the revised manuscrit (strictly define the lines in the revised manuscript).
Author Response
Concern 1: Amp B is a poorly soluble drug, and it should be added together with phospholipids and cholesterol during liposomes preparation, not during film hydration phase as indicated by the authors. These are the basics of liposomes and liposomes preparation. Moreover, the authors must indicate the amount of Amp B added to form the Amp-liposomes. The authors indicated the amounts of the drug and markers for in vitro release and in vivo studies.
Since the drug is not soluble in water, ultracentrifugation is not appropriate method for separation of the non-entrapped drug as the drug will precipitate together with the liposomes. Instead, gel-filtration method based on the use of Sepharose CL-4B or minicolumn centrifugation method (Sephadex) should be used. These methods are accurate for all the drugs, while ultracentrifugation is suitable only for hydrophilic drugs.
Although AmpB is soluble in DMSO, after addition of water (higher amount), it will start to precipitate. The encapsulation efficiency is, therefore, false positive. It could be so high but only if the drug is added together with the lipid phase and by selecting the appropriate lipids (please check the composition of commercially available amphotericin B liposomes).
Moreover, the authors should be aware that certain amount of DMSO is still present in liposomes. Such formulation is not appropriate for further human use. The above answer clarify this point as well.
I suggest the authors to prepare the Amp B-liposomes with adding the drug together with the phospholipid and cholesterol, which is a regular way of preparation liposomes with lipophilic drugs. The encapsulation efficiency determination should be checked again, but with using gel-column filtration/chromatography method using Sepharose CL-4B or minicolumn centrifugation method (using appropriate Sephadex).
Response 1: These queries are valid. Here, we are clarifying the points raised by the erudite reviewer and we incorporated in the text as suggested.
AmpB was dissolved in 2-3 ml of methanol and added to the mixture of chloroform and methanol (9:1) with previously dissolved egg PC and cholesterol. The mixture was evaporated in a rotary evaporator to obtain a thin homogenous film in a rotary flask in case of their entrapment in liposomes. The film was hydrated in normal saline in a bath sonicator for the formation of liposomes.
Amp B was dissolved in as minimum of DMSO as possible and diluted with water only in case of their entrapment in plasma beads. We were aware of the precipitation issues which we faced during optimizing our system, as also pointed out by the erudite reviewer.
We have corrected this in the text : lines 127-132 & lines 167-168.
Since we had a high entrapment of Amp B, volume was not a problem. And since the exact volume would vary batch to batch, we kept the volume of the liposomes constant (20 µl/100µl of the plasma, as mentioned in the text); we diluted the stock liposomal formulations accordingly.
Concern 2: The authors’ explanation why were in vitro release experiments performed at 25 °C, not at 37 °C, must be denoted in the manuscript. The authors must indicate in the revised manuscript that 25 °C is not appropriate temperature.
Response 2: The erudite reviewer is right in pointing out that the in vitro study should be performed at 37 °C, but our laboratory performed the in vitro release at 25 °C (Ahmad et al., 2011). We have mentioned this in the manuscript, lines 164-165.
Concern 3: Despite of the cited reference, the lamellarity of the liposomes was not checked and TEM figures are not provided. The used terms LUVs and MLVs are hence not appropriate. Instead, only term liposomes, but indicating the entrapped compound, e.g., Amp B-liposomes, should be used. Please re-write throughout the manuscript.
Response 3: We have incorporated the erudite reviewer's suggestion throughout the text and replaced the MLVs and LUVs with liposomes. They can be seen as edits in blue throughout the manuscript (Lines 123, 133, 141, 253, 272, 391, 409, 411, 415, 423)
Concern 4: In response to the reviewer, the authors must specify exactly what change is and where the change is made in the revised manuscript (strictly define the lines in the revised manuscript).
Response 4: We have done as suggested. Thank you.
Reviewer 3 Report
1) Response 1. We have described the mechanism of Amp B distribution in liposome in the discussion section (line 377-380).
Lines 378-385 do not say a word about AmpB distribution in the systems studied by the authors. Where is the medicine? Associated with the surface? Inside the lipid layer? Reference 47, which is given in this paragraph, is devoted to working with microparticles from 2x to 40 microns, which is definitely more than liposomes. Work 48 discusses, firstly, a different drug, and secondly, a different approach to drug delivery. It seems that the entire literary review was compiled only by keywords. I do not understand the relevance of citing these works.
2) The main statistically reliable method of liposome characterization is NTA (analysis of nanoparticle trajectories). Is there any data in the literature on your commercial systems by this method.
3) The obvious relevance of the work for people in the working group/institute/field of interest is not always obvious to a wide range of researchers. In response to the question, the authors cite references to the works of the last decade (2011-2012). The question of relevance remains open.
Author Response
Concern 1: Lines 378-385 do not say a word about AmpB distribution in the systems studied by the authors. Where is the medicine? Associated with the surface? Inside the lipid layer? Reference 47, which is given in this paragraph, is devoted to working with microparticles from 2x to 40 microns, which is definitely more than liposomes. Work 48 discusses, firstly, a different drug, and secondly, a different approach to drug delivery. It seems that the entire literary review was compiled only by keywords. I do not understand the relevance of citing these works.
Comment: We investigated the overall entrapment drug/ molecules of interest and release of the drugs in in vitro and in vivo in the various systems. The fibrin based plasma bead system was porous, so we wanted to entrap drugs more effectively and sustain its release. We believe that the any entrapped moiety in general is entrapped in the liposomes and diffuses from it, and further being obstructed by the fibrin network causes gradual leakage of the drug. In case of Amp B, the drug is entrapped in the liposomes to improve its retention. It also binds to the plasma albumin, causing a marked slow release.
This is an established fact that hydrophobic molecules entrapped in the hydrophobic core and hydrophilic molecules would entrapped in the aqueous core of the lipid bilayer of the liposome, so the distribution of the drug will follow the obvious rule. We are not unravelling a novel lipid based delivery system. Our motive is to improve overall entrapment and controlled release pattern via the plasma bead system
.
Concern 2: The main statistically reliable method of liposome characterization is NTA (analysis of nanoparticle trajectories). Is there any data in the literature on your commercial systems by this method.
Response 2: We did not perform detailed liposomal characterization. Liposomes have fairly been characterized and commercially available [Amp B encapsulated liposome (AmBisome®; LAmB)] are in the market. Our focus was to achieve a dual entrapment, with plasma being the unique focus so as to utilize its potential as a drug delivery system, while trying to overcome the apparent drawback concerning its porosity.
Concern 3: The obvious relevance of the work for people in the working group/institute/field of interest is not always obvious to a wide range of researchers. In response to the question, the authors cite references to the works of the last decade (2011-2012). The question of relevance remains open.
Response 3: This study proposes a fibrin-based plasma beaded system as a potential drug delivery system. As also mentioned in the discussion, the beads can be prepared from autologous plasma, and entrapped with the molecule of interest and reinjected into the system, utilizing the use of no external agent except CaCl2. The laboratory setup is simple. We have also developed a system for antigen delivery and investigated the potential of plasma beads in combination with erythrocytes with encouraging results (Fatima et al., 2018). We believe this system is a simple, yet safe and efficient for drug and antigen delivery, and can be used in future for personalized medicine approaches.
Reviewer 4 Report
It's ok for publishing now.
Author Response
No comments